# On the Origins of Omicron’s Unique Spike Gene Insertion

**DOI:** 10.3390/vaccines10091509

**Published:** 2022-09-09

**Authors:** A. J. Venkatakrishnan, Praveen Anand, Patrick J. Lenehan, Rohit Suratekar, Bharathwaj Raghunathan, Michiel J. M. Niesen, Venky Soundararajan

**Affiliations:** 1nference, Cambridge, MA 02139, USA; 2nference Labs, Bengaluru 560017, Karnataka, India; 3nference, Toronto, ON M5V 1M1, Canada

**Keywords:** COVID-19, Omicron, template switching

## Abstract

The emergence of a heavily mutated SARS-CoV-2 variant (Omicron; Pango lineage B.1.1.529 and BA sublineages) and its rapid spread to over 75 countries raised a global public health alarm. Characterizing the mutational profile of Omicron is necessary to interpret its clinical phenotypes which are shared with or distinctive from those of other SARS-CoV-2 variants. We compared the mutations of the initially circulating Omicron variant (now known as BA.1) with prior variants of concern (Alpha, Beta, Gamma, and Delta), variants of interest (Lambda, Mu, Eta, Iota, and Kappa), and ~1500 SARS-CoV-2 lineages constituting ~5.8 million SARS-CoV-2 genomes. Omicron’s Spike protein harbors 26 amino acid mutations (23 substitutions, 2 deletions, and 1 insertion) that are distinct compared to other variants of concern. While the substitution and deletion mutations appeared in previous SARS-CoV-2 lineages, the insertion mutation (ins214EPE) was not previously observed in any other SARS-CoV-2 lineage. Here, we consider and discuss various mechanisms through which the nucleotide sequence encoding for ins214EPE could have been acquired, including local duplication, polymerase slippage, and template switching. Although we are not able to definitively determine the mechanism, we highlight the plausibility of template switching. Analysis of the homology of the inserted nucleotide sequence and flanking regions suggests that this template-switching event could have involved the genomes of SARS-CoV-2 variants (e.g., the B.1.1 strain), other human coronaviruses that infect the same host cells as SARS-CoV-2 (e.g., HCoV-OC43 or HCoV-229E), or a human transcript expressed in a host cell that was infected by the Omicron precursor.

## 1. Introduction

A new SARS-CoV-2 variant with an extensively mutated Spike protein was first reported to the World Health Organization (WHO) from South Africa on 24 November 2021, with the first sample collected on 9 November 2021. This strain was subsequently denoted as the Omicron variant (WHO nomenclature) and B.1.1.529 (Pango lineage) [1]. The rapid assessment of the variant by *The Technical Advisory Group on SARS-CoV-2 Virus Evolution* and classification of Omicron as a variant of concern by the WHO within 48 h facilitated timely epidemiological surveillance. After its initial discovery, this variant rapidly spread across the globe and was detected in over 75 countries across 6 continents by 16 December 2021 [2,3]. After this, multiple Omicron sublineages emerged (Pango lineages BA.1, BA.2, BA.3, BA.4, BA.5, and descendants thereof) and drove case surges around the world. 

Thoroughly characterizing the mutational profile of Omicron is a necessary step to interpret its shared or distinctive clinical phenotypes with respect to other variants, its sensitivity or resistance to existing vaccines, and whether Omicron-like variants that evolve in the future may have heightened virulence. Indeed, SARS-CoV-2 has evolved into different variants of concern and variants of interest through a combination of missense, deletion, and insertion mutations. For example, the D614G substitution in the Spike (S) protein, which emerged early and has been detected in nearly all SARS-CoV-2 genomes in GISAID since mid-2020, increases the replication capacity and infectivity of SARS-CoV-2 [4,5]. Other substitutions (e.g., E484K and E484A) have led to significant changes in the Spike–ACE2 binding affinity, and deletions (e.g., ΔY144) have modulated the effects of neutralizing anti-Spike antibodies [6,7,8,9,10,11,12,13]. Insertion mutations have been less prevalent in the evolution of SARS-CoV-2 [14]. However, one of the most functionally consequential mutations in the evolutionary history of SARS-CoV-2 to date was the “PRRA” Spike protein insertion in the S1/S2 cleavage site, which introduced the polybasic furin cleavage site that mimics the RRARSVAS peptide in human ENaC-alpha [15,16,17,18,19]. This insertion plays an important role in the transmission of SARS-CoV-2, at least in part by facilitating an endosome-independent entry pathway into respiratory epithelial cells that bypasses important innate antiviral responses [18]. It is also mechanistically required for the syncytium-mediated death of lymphocytes, which may contribute to the lymphopenia that is often clinically observed in COVID-19 patients [20,21,22,23]. The availability of 5.8 million SARS-CoV-2 genomes covering ~1500 lineages from over 203 countries/territories in the GISAID database since the beginning of the pandemic provides an opportunity to characterize the genomic landscape of the Omicron variant in comparison to other SARS-CoV-2 variants.

In this study, we compare the mutational profiles of early Omicron genomes (primarily sublineage BA.1, hereafter referred to as “Omicron”) with all other SARS-CoV-2 lineages, including the variants of concern and variants of interest. We highlight that Omicron’s Spike protein harbored an insertion mutation ins214EPE that was absent in all other SARS-CoV-2 lineages at the time of its emergence. Given the salience of viral genetic recombination and the debated plausibility of host genome integration by SARS-CoV-2 [24,25,26], we considered a variety of host–viral and interviral genomic matter exchange scenarios that may have contributed to the adoption of this insertion mutation in the precursor variant of Omicron. We discuss potential sources for the origin of ins214EPE and highlight the need to experimentally characterize the role of ins214EPE in viral transmission and immune evasion.

## 2. Methods

### 2.1. Analysis of Mutations Defining the Omicron Lineage

Core mutations were derived for parental lineages from the Coronavirus Antiviral Research Database (CoV-RDB; covdb.standford.edu; accessed 10 December 2021) for each variant of interest or variant of concern: Alpha (B.1.1.7), Beta (B.1.351), Gamma (P.1), Delta (B.1.617.2), Lambda (C.37), Mu (B.1.621), Eta (B.1.525), Iota (B.1.526), Kappa (B.1.617.1), and Omicron (BA.1) [27]. All SARS-CoV-2 genome sequences corresponding to the Omicron variant were directly derived from the GISAID database [2,3]. There were 1448 SARS-CoV-2 genomes annotated as B.1.1.529/BA.1/BA.2 as of 13 December 2021. These genomes were compared to other (non-Omicron) genomes deposited in GISAID before the same date to determine the number of lineages in which each Omicron Spike protein mutation was previously observed.

To understand which SARS-CoV-2 mutations correlate with COVID-19 test positivity, we identified “surge-associated mutations” using genomic data from GISAID (5,781,715 sequences from 203 countries/territories between December 2019 and December 2021) and epidemiology data from Our World in Data (OWID) [28], as described in our previous study [29]. A mutation is considered to be “surge associated” if it satisfies the following criteria: (1) it is present in at least 100 SARS-CoV-2 sequences in GISAID; and (2) in a period of three consecutive months during which there was a monotonic increase in PCR positivity by at least 5% in a given country, the prevalence of the mutation in that country also monotonically increased by at least 5%.

To assess the prevalence of ins214EPE in each Omicron lineage (B.1.1.529 and all BA sublineages), SARS-CoV-2 genomes (n = 11,945,950) were collected from GISAID on 18 July 2022 along with their annotated metadata, including lineage assignment and mutational profiles. To identify other insertions in this region that were observed prior to the emergence of Omicron, we filtered these 11,945,950 genomes to those meeting the following criteria: (i) high coverage, (ii) complete sequence, (iii) collection date on or before 13 December 2020 (i.e., the data cutoff date for our initial analysis of Omicron mutations), (iv) not assigned to the Omicron lineage (B.1.1.529 or a BA sublineage), (v) containing an insertion between positions 210 and 218 of the Spike protein, (vi) the inserted sequence does not contain a stop codon or a low-confidence alignment (i.e., amino acid indicated as “X”), and (vii) the insertion is not ins214EPE.

### 2.2. Nucleotide 9-mer Search to Identify Candidate Viral and Human Templates for ins214EPE

An exact 9-mer search for all three possible inserts (5′-AGCCAGAAG-3′, 5′-GAGCCAGAA-3′, and 5′-GCCAGAAGA-3′) and their reverse-complement sequences was performed across three different databases: (i) the human transcriptome, (ii) SARS-CoV-2 genomes from GISAID, and (iii) human-infecting *Coronaviridae* family viruses. For a reference Omicron genome (EPI_ISL_6640916) [30], we also performed a modified search to identify all nine-nucleotide sequences that differed from the possible insertion sequences by only one nucleotide (i.e., allowing for a single mismatch). The human transcript sequences (n = 244,939) were downloaded from the GENCODE database [31] (version 39; GRCh38.p13 of the human genome). Coding sequences (CDSs) for 5,781,715 SARS-CoV-2 genomes were accessed from GISAID [2] on 13 December 2021. All available sequences from human-infecting *Coronaviridae* family (taxid:11118) viruses were accessed from the NCBI virus database (https://www.ncbi.nlm.nih.gov/labs/virus, accessed on 13 December 2021; n = 574,178 complete genomes with 4,028,478 CDSs) [32]. In the final analysis, we excluded genomes that were labeled as SARS-CoV, SARS-CoV-2, or Middle East respiratory syndrome (MERS)-CoV (or otherwise named strains of these viruses) so as to consider only seasonal and enteric coronavirus strains—human coronavirus (HCoV)-229E, HCoV-OC43, HCoV-NL63, and human enteric coronavirus strain 4408.

### 2.3. Assessment of Homology between Regions Flanking Insertion and Origin Sites

For a given insertion, we defined a putative origin as the matched sequence in a viral genome, a viral anti-genome, or a human transcript. We then assessed the homology between the 35 nucleotides upstream of the insertion and the 35 nucleotides upstream of the origin, and similarly we assessed the homology between the 35 nucleotides downstream of the insertion and the 35 nucleotides downstream of the origin. To assess the similarity between any given pair of nucleotide sequences, we defined a function using the Bio.pairwise2 module from Biopython v1.76 (https://biopython.org/wiki/Documentation accessed on 10 January 2022), which performs a global alignment of nucleotide sequences using a custom scoring scheme (i.e., +5 for a match, −4 for a mismatch, 0 for gap start and extension). This score ranges from 0 (no matches) to 175 (perfect match for all 35 nucleotides). We further defined a normalized homology score (NHS), ranging from 0 to 100, which is calculated by simply dividing the homology score by 175 and multiplying the result by 100. We also applied a similar protocol to assess the homology between shorter upstream and downstream sequences (7 nucleotides), where the NHS was calculated by dividing the homology score by 35 and multiplying the result by 100.

A set of “positive control” template-switch-mediated insertions was obtained from Appendix A from the prior analysis by Garushyants et al. [14]. We filtered this table to include only those insertions that were assigned a mechanism of “Template switch” and which were 12 or more nucleotides long. This set of insertions is shown in Appendix A. For each insertion, the genomic positions of the origin and insertion sequences are provided in this table, along with the Pango lineage assigned to the genome(s) harboring the insertion. We searched SARS-CoV-2 genomes in GISAID to identify those genomes that contained the provided origin and insertion sequences at approximately the provided genomic positions. The GISAID identifiers for the corresponding insertion-containing genomes are provided in Appendix A. For each of these genomes, we then obtained the 35 nucleotide sequences upstream and downstream of the insertion and origin. Finally, we calculated the NHS for each relevant pair of sequences: (i) the 35 nucleotides upstream of the insertion versus the 35 nucleotides upstream of the origin, and (ii) the 35 nucleotides downstream of the insertion versus the 35 nucleotides downstream of the origin. These scores, and the local alignments contributing to them, are shown for each individual genome in Appendix A.

As a “negative control” analysis, we calculated the NHSs between 10,000 pairs of randomly selected non-overlapping *n*-mers (35-mers or 7-mers) from the original SARS-CoV-2 genome (NC_0545512.2; https://www.ncbi.nlm.nih.gov/nuccore/NC_045512, accessed on 10 January 2022). To generate each pair of *n*-mers, we randomly selected two genomic coordinates (i.e., nucleotide positions between 1 and 29,903) as the starting positions for an *n*-mer nucleotide sequence. If the two positions were within *n* nucleotides of one another, a new pair of coordinates was selected because these would, by definition, generate overlapping *n*-mers. Similarly, if either position was less than *n* nucleotides from the end of the SARS-CoV-2 genome sequence, then a new pair of coordinates was selected.

### 2.4. Single-Cell Analysis of Coronavirus Receptor Co-Expression

Publicly available single-cell RNA sequencing datasets were obtained and processed as described previously [33,34]. The data are hosted at https://academia.nferx.com/dv/202011/singlecell/, accessed on 10 January 2022, and include approximately 2.8 million cells derived from dozens of independent studies covering most major human tissues. We determined the total numbers and percentages of cells (of these 2.8 million total cells) expressing each gene encoding a coronavirus receptor of interest (e.g., ACE2, ANPEP, DPP4), along with the numbers and percentages of cells co-expressing ACE2 and ANPEP or ACE2 and DPP4. We also performed similar analyses on two specific studies of interest: (i) a study of nasopharyngeal and bronchial samples from COVID-19 patients and healthy controls (approximately 33,000 cells) [35], and (ii) a study of ileal biopsies from Crohn’s disease patients (approximately 136,000 cells) [36]. For these studies, we also evaluated co-expression in the cell types that showed the strongest expression of these genes. To evaluate whether the degree of co-expression for ACE2 and ANPEP or DPP4 was more than expected by chance, we calculated the observed-to-expected ratio of co-expression, assuming that the expression of each gene is distributed randomly across the analyzed cells. Specifically, this value was calculated by dividing the co-expressing percentage by the product of the individual expression percentages and multiplying the result by 100.

## 3. Results

### 3.1. Comparison of Mutations in Omicron to Previous SARS-CoV-2 Lineages Shows the Presence of a Unique Insertion Mutation in Omicron’s Spike Protein 

Omicron harbors 37 mutations in the Spike protein, which include 6 deletion mutations, 1 insertion mutation, and 30 substitution mutations [27]. Of the 37 mutations, 16 were associated with regional case surges prior to the Omicron era (Appendix A; see Methods) [29]. Comparing these Spike protein mutations in Omicron with pre-existing variants of concern (VOCs: Alpha, Beta, Gamma, and Delta) shows that 26 mutations are distinct to Omicron, while the remaining 11 are shared with at least one other VOC (Figure 1A, Appendix A). When compared to the Delta variant, the mutational load of Omicron is particularly high in the Spike protein sequence, with more similar rates of mutation in regions of the viral genome encoding other proteins (Appendix A). 

We next analyzed which of these 26 mutations (i.e., Spike mutations present in Omicron but not in any prior VOCs) appeared in the prior variants of interest (VOIs: Lambda, Mu, Eta, Iota, and Kappa) or other prior SARS-CoV-2 lineages by comparing them with mutations from 5,781,715 genomes corresponding to ~1500 lineages from the GISAID database. Two of these mutations were present in VOIs (A67V in Eta, T95I in Mu and Iota) (Appendix A), while twenty-three others appeared in previously collected genomes assigned to SARS-CoV-2 lineages that were not classified as VOIs or VOCs (Figure 1B, Appendix A). Interestingly, only the insertion mutation ins214EPE had not been previously observed in any SARS-CoV-2 lineages (Figure 1B, Appendix A), although other insertions at or near this position had been observed in several lineages before the emergence of Omicron (Appendix A) [14,37]. Specifically, of the 1168 SARS-CoV-2 genomes in GISAID harboring this insertion at the time of this analysis, 1164 were classified as Omicron. Of the remaining four genomes, three had not yet been assigned a Pango lineage, while the other one (which was deposited on 24 November 2021) was labeled as B.1 (a parent lineage of Omicron). 

The EPE insertion (ins214EPE) on Omicron maps to the Spike protein’s N-terminal domain (NTD) distal from the antibody-binding supersite [11]. However, the loop where the insertion is present maps to a known human T-cell epitope on SARS-CoV-2 [38]. Further studies will be necessary to understand whether this insertion may help SARS-CoV-2 to escape T-cell immunity [14]. A recent study also suggests that insertions in the NTD, including ins214EPE in Omicron, may increase viral transmissibility by enhancing sialic acid receptor binding [39]. Such a functional consequence may be consistent with the observation that multiple SARS-CoV-2 lineages have acquired insertions of 1–6 additional amino acids at or near this same site (i.e., between positions 210 and 218) (Appendix A) [37]. Given the potential for insertions to impact SARS-CoV-2’s virulence (e.g., the PRRA insertion giving rise to a polybasic furin cleavage site in the original SARS-CoV-2 strain), it is important to understand the functional significance and evolutionary origins of ins214EPE in the Omicron variant [15,16,17,19]. 

It should be noted that BA.1 was the predominant Omicron sublineage at the time of these initial characterizations in December 2021. Multiple other sublineages (BA.2, BA.3, BA.4, and BA.5) sequentially emerged, and became the dominant (i.e., most prevalent) strains of SARS-CoV-2. While ins214EPE is present in over 75% of BA.1 sequences collected in GISAID as of July 2022, it is almost uniformly absent from all of the subsequent sublineages (Appendix A). In the remainder of this perspective, we consider possible mechanisms that could have led to the generation of this insertion, which was a defining feature of the earliest circulating Omicron variant, BA.1.

### 3.2. Template Switching Is a Plausible Mechanism for the Origin of ins214EPE in Omicron

Ins214EPE is an in-frame insertion of nine nucleotides that occurs between nucleotide positions 22204 and 22207. Based on the local sequence alignment, there are three candidate insertions: (1) 5′-GAGCCAGAA-3′ between 22204 and 22205, (2) 5′-AGCCAGAAG-3′ between 22205 and 22206, and (3) 5′-AGCCAGAAG-3′ between positions 22206 and 22207 (Appendix A). According to a recent analysis of the secondary structure of the reference SARS-CoV-2 genome, the nucleotides between positions 22205 and 22208 (5′-GAUC-3′) constitute an RNA loop [40], which is notable given that RNA loops appear to be more prone to insertions than stems.

A previous analysis of sequences deposited in GISAID concluded that insertions in the SARS-CoV-2 genome most likely arise from one of three mechanisms: (1) local duplications, (2) polymerase slippage, or (3) template switching [14]. While local duplications were found to explain some short (fewer than nine nucleotides) and long (nine or more nucleotides) insertions, the observation that these groups had distinct nucleotide compositions, phyletic patterns, and genomic localizations led to the conclusion that short and long insertions typically arise from distinct mechanisms [14]. Specifically, it was suggested that polymerase slippage is the best explanation for most short insertions, which typically have an excess of uracil nucleotides and are non-monophyletic. It is proposed that slippage is most likely to occur during runs of uracils, owing to the slow processing of polyU tracts that has been demonstrated for the RNA-dependent RNA polymerase (RdRp) of SARS-CoV-1 and is hypothesized to be true for the RdRp of SARS-CoV-2 as well [14,41]. On the other hand, it was posited that long insertions typically arise from template switching, given that their nucleotide composition is consistent with the SARS-CoV-2 genome, they are typically monophyletic, and they tend to occur at or near sites that were previously identified as hotspots for template switching [14,42]. It is notable that template switching is a normal part of the life cycle for *Coronaviridae*, as discontinuous transcription via template switching is responsible for the synthesis of subgenomic RNAs (sgRNAs) [43,44]. In this light, we asked which mechanism is the most fitting explanation for the origin of ins214EPE in Omicron. 

As mentioned above, exact duplication of an adjacent nucleotide sequence has been observed previously as a mechanism for both short and long insertions in the SARS-CoV-2 genome [14]. For example, in three GISAID sequences of the B.1.429 lineage, there is a duplication of 24 nucleotides (5′-AAAAGAAGAAGGCTGATGAAACTC-3′) resulting in the sequence 5′-*AAAAGAAGAAGGCTGATGAAACTC***AAAAGAAGAAGGCTGATGAAACTC**-3′ at position 29387 (corresponding to the nucleocapsid protein amino acid position 372) [14]. However, the inserted nucleotide sequence resulting in the Omicron ins214EPE is not a result of such a local duplication, as it is not identical or closely homologous to the preceding or subsequent nucleotide sequences in the original reference genome sequence of SARS-CoV-2, nor to that of the Omicron variant (Appendix A). 

We thus asked whether polymerase slippage or template switching was a more plausible explanation for the origin of ins214EPE. For several reasons, it appears that template switching is the more plausible hypothesis. First, this is a long insertion per the definition described previously (nine or more nucleotides), and long insertions are more likely to arise from template switching than from polymerase slippage [14]. That said, we recognize that with exactly nine nucleotides, this is a borderline case between short and long insertions, and so its length alone may have limited value in distinguishing between these mechanisms. Second, this insertion has no uracil nucleotides, contrary to the expected excess uracils in slippage-mediated insertions [14]. Third, this insertion is monophyletic, although it is worth noting that other insertions at the same location have been observed in several other SARS-CoV-2 lineages (Appendix A) [14,37]. Finally, the insertion occurs near previously described sites of potential non-canonical template switching. Specifically, there were non-canonical junctions observed 30 nucleotides upstream and 60 nucleotides downstream of this site (at positions 22183 and 22276, respectively) [42]. 

### 3.3. Candidate Templates for the Origin of ins214EPE in Omicron

While it is not certain that ins214EPE was generated by template switching, the points above illustrate that this is a plausible mechanism. If true, it would be of interest to determine candidate template RNA molecules from which this insertion could have arisen. We reasoned that there are three broad categories of most likely templates: (1) genomic material of SARS-CoV-2 itself (i.e., the positive-sense genomic RNA or the negative-sense anti-genomic RNA); (2) genomic or anti-genomic material of other viruses that have the capacity to co-infect the same cells as SARS-CoV-2; and (3) human transcripts that are expressed in cells infected with SARS-CoV-2 (Figure 2A,B). Although the latter category is not a well-described method of template switching [45], it has been suggested previously that insertions in SARS-CoV-2 genomes could be derived from the host transcriptome [46]. We identified exact matches for the forward and/or reverse-complement sequences in all three categories (Table 1, Figure 2C). 

There are no exact matches in any Omicron genomes collected to date for the three candidate sequences outside of the insertion site itself. This is notable because in previous template-switch-mediated insertions in the SARS-CoV-2 genome, the putative insertion template (“origin”) has typically been present in the insertion-containing genome (Appendix A) [14]. For the Omicron insertion, on the other hand, the only SARS-CoV-2 sequences in GISAID containing exact forward or reverse-complement matches are assigned to other SARS-CoV-2 lineages (or were not assigned to any lineage at the time of this analysis). There are several possible implications or interpretations of this finding.

First, substitutions can be introduced within the inserted sequence during template switching itself, or during subsequent rounds of viral replication, which would result in imperfect matching between the insertion and template sequences. This is particularly relevant for Omicron, which represents a long phylogenetic branch that arose after presumably several months of unobserved evolution [30,47]. We indeed found that the reference Omicron genome (EPI_ISL_6640916) [30] harbors several nucleotide 9-mers that differ from the candidate insertions by only a single nucleotide (Appendix A), and additional 9-mers that differ by two or three nucleotides. These should be considered as possible templates for ins214EPE. Second, it is possible that the utilized template was derived from a co-infecting SARS-CoV-2 variant that does harbor the exact inserted sequence. Recombination between SARS-CoV-2 lineages in the context of simultaneous co-infection has been described previously, with particularly high recombination rates seen in the Spike protein sequence [43,48]. The distributions of lineages comprising the matched genomes for each candidate insertion are shown in Appendix A. Finally, we noticed that several of the non-Omicron genomes with exact matches to one or more of the candidate insertions have been assigned to the B.1.1 Pango lineage (or sublineages thereof) (Appendix A). Given that Omicron (originally Pango lineage B.1.1.529) is a phylogenetic descendant of B.1.1, this suggests that the ancestral genome that evolved into Omicron could have provided the necessary template for this insertion.

We also identified several genomes (or anti-genomes) of seasonal or enteric human-infecting coronaviruses that contain one or more of the putative insertion sequences. For example, the genomes of multiple human coronavirus OC43 (HCoV-OC43) and human enteric coronavirus strain 4408 both contain 5′-GCCAGAAGA-3′ and 5′-GAGCCAGAA-3′ in their nucleocapsid and replicase polyprotein genes, respectively (Appendix A). Furthermore, 5′-GAGCCAGAA-3′ was present in 33 HCoV-229E anti-genomes, and 5′-GCCAGAAGA-3′ was present in 2 HCoV-NL63 anti-genomes (Appendix A). The importance of recombination between coronaviruses has been highlighted recently [49], and its potential is supported by clinical reports showing that COVID-19 patients are co-infected with other respiratory pathogens, including non-SARS-CoV-2 viruses of the *Coronaviridae* family, at relatively high frequencies [50,51,52]. Furthermore, host receptors utilized by other coronaviruses (e.g., ANPEP and DPP4) are co-expressed with the SARS-CoV-2 receptor (ACE2) at the single-cell level in respiratory and/or gastrointestinal epithelial cells (Appendix A) [33], which could facilitate co-infection at the cellular level (a prerequisite for genomic recombination). Intestinal co-expression is relevant given the evidence that SARS-CoV-2 and other coronaviruses can infect enterocytes [53,54,55]. That said, it is worth noting that if template switching was indeed the mechanism by which this insertion arose, it is possible that the genomic material of respiratory pathogens outside of the *Coronaviridae* family (e.g., influenza, respiratory syncytial virus, parainfluenza, human metapneumovirus) could also serve as substrates in the context of such cellular co-infection. Genetic recombination between co-infecting viruses has been described, but this is more likely to occur between viruses in the same family with a high degree of genomic homology [56,57,58].

Finally, there were 4677 human transcripts (from 1534 genes) containing the forward sequence 5′-GAGCCAGAA-3′, and 3264 human transcripts (from 1220 genes) containing the reverse-complement sequence. Similar summary statistics are shown for the two other potentially inserted nine-nucleotide sequences in Table 1.

### 3.4. Consideration of Local Homology for the Candidate Templates

This landscape of possible templates, particularly among human transcripts, is expectedly quite vast given the total space of possible 9-mer nucleotide combinations (4^9^ = 262,144). This raises the question of whether the most likely candidates may be those transcripts that have more homology surrounding the inserted sequence, as complementary base pairing resulting from such local similarity can increase the likelihood of serving as a template for recombination [45]. Indeed, in the normal process of coronavirus genomic replication, the prevailing model is that subgenomic RNAs are generated during negative-strand synthesis via a template-switching mechanism that relies on homology between conserved transcription regulatory sequence (TRS) elements dispersed strategically throughout the genome [42,44,59,60]. However, in the context of SARS-CoV-2, non-canonical transcripts with junctions that are not derived from TRS sequences and that share little homology between the 5′ and 3′ sites suggest that template switching guided by partial complementarity or other mechanisms may also play a role [42,61].

To study the homology between the regions surrounding template-switch-mediated insertions and their origins (i.e., the putative template that was copied to generate the insertion), we first considered a “positive control” set of four 12–15-nucleotide insertions in SARS-CoV-2 that were previously attributed to template switching with high confidence (Appendix A) [14]. We calculated a normalized homology score (NHS) between the 7 or 35 nucleotides upstream or downstream of the insertion and origin sequences in these genomes, respectively (see Methods). Surprisingly, the degree of homology observed was generally not higher than expected by chance, as assessed via the NHS distribution of 10,000 randomly paired non-overlapping 7-mer or 35-mer nucleotide sequences from the SARS-CoV-2 genome (Appendix A, Appendix A). This suggests that local homology may not be a prerequisite for the generation of template-switch-mediated insertions in SARS-CoV-2 [42].

Nevertheless, we still assessed the homology between the 35 nucleotides upstream or downstream of the Omicron insertion and the 35 nucleotides upstream or downstream of all candidate templates (see Methods). The NHS distributions for these candidates were generally similar to the distributions observed previously for randomly selected SARS-CoV-2 35-mers (Appendix A). Candidates in each category with the highest degrees of homology in the flanking upstream or downstream sequences included SARS-CoV-2 genomes from the lineages B.1.609 (NHS = 69) and AY.103 (NHS = 66), the HCoV-229E Spike protein (NHS = 63), and human transcripts of ACTN1 (NHS = 74) and EMC4 (NHS = 71). Some candidate templates also showed more homology in shorter sequences directly upstream and/or downstream of the inserted sequence. For example, in the reverse complement of the human TMEM245 transcript, there is a 17-nucleotide stretch with exact homology to the insertion-containing region of the Omicron genome (i.e., the nine-nucleotide inserted sequence plus eight exactly matched flanking nucleotides) (Appendix A).

## 4. Discussion

Omicron is more highly transmissible than prior variants [62,63], is less susceptible to neutralization by monoclonal antibodies and sera of vaccinated individuals [64,65,66,67,68,69,70], and is more likely to cause re-infections and vaccine breakthrough infections [71,72]. Among the many mutations in its Spike protein, ins214EPE is the only one that was not observed in other lineages prior to the emergence of Omicron. Whether this insertion, alone or in concert with other mutations, contributes to heightened transmissibility or lower susceptibility to neutralization by antibodies warrants further investigation.

While we cannot definitively determine the mechanism that gave rise to ins214EPE, we propose that template switching is a plausible explanation (Figure 2). Although the RNA-dependent RNA polymerases of SARS-CoV-2 and other coronaviruses do normally utilize template switching to generate subgenomic RNAs [42,59], it appears that template-switch-mediated insertions may derive from a non-canonical form of this process, in which a high degree of local homology (e.g., homologous TRS core sequences in the leader and body regions of the genome) is not essential [42,61]. It is not clear why a given sequence would be utilized as a template in the absence of local homology, but this mechanism of “template selection” could involve secondary RNA structures or other unappreciated aspects of the SARS-CoV-2 replication machinery [73]. Here, we highlight several possible sources of the template for this insertion, including the SARS-CoV-2 genome itself, along with the genomes of other viruses or human transcripts. The use of SARS-CoV-2 genomic material as the template is supported by the finding that SARS-CoV-2 genome replication occurs in organelles that spatially concentrate the viral genomic material and replication machinery [74]. That said, it might indeed be possible for non-SARS-CoV-2 viral genomes or host mRNAs to be aberrantly included in these organelles, rendering them accessible for utilization during template switching as well.

There may be additional mechanisms that contribute to the acquisition of insertions by SARS-CoV-2 beyond those considered here. Saltational viral evolution in immunocompromised patients has been suggested to underlie the emergence of highly mutated variants [75], and it is possible that ins214EPE (along with other Omicron-defining mutations) emerged in this context. Such individuals may be more prone to simultaneous co-infection with multiple SARS-CoV-2 variants, or with SARS-CoV-2 and other respiratory pathogens. It is also evident that the evolution of SARS-CoV-2 can occur in non-human species such as mice, deer, and mink or other mustelids [76,77,78,79], in which case other viral genomes and transcripts should be considered as possible templates. A recent analysis suggests that the proofreading exoribonuclease (encoded in nonstructural protein 14, or nsp14) is required for at least some of the genetic recombination observed in SARS-CoV-2 [80], but the potential mechanisms described here do not account for this. Finally, it is reasonable to question whether there is a relationship between ins214EPE and the three-nucleotide deletion (ΔN211) that occurs shortly upstream of it in most Omicron BA.1 sequences. However, because most sequences in GISAID with other insertions at position 214 do not possess such neighboring deletions (Appendix A), we believe that this proximal deletion was not a mechanistic prerequisite for the generation of the Omicron insertion.

It is not clear whether any one of these insertion-generating mechanisms would have more far-reaching consequences than the others. Template switching offers the intriguing possibility of new SARS-CoV-2 lineages borrowing protein domains or subdomains from previous variants, other viruses, or human proteins. However, it is worth noting that several of the possible templates that we identified for ins214EPE were derived from viral anti-genomes (i.e., from HCoV-229E or SARS-CoV-2 anti-genomes) or the reverse-complement sequences of human transcripts (e.g., TMEM245). In such cases, the polypeptide encoded in the Omicron genome would differ from that encoded by the positive-sense strand of the template. The genomic location and/or amino acid content of the inserted sequence may be more important than its origin, and we thus highlight the need to characterize the functional impact of ins214EPE on the clinical and epidemiological properties of the Omicron variant. Importantly, the data and hypotheses presented here are not sufficient to make inferences about properties such as transmissibility, immune evasion, or disease severity. Even if Omicron did acquire an insertion by utilizing a host transcript or the genome of a common-cold-causing coronavirus (e.g., HCoV-OC43, HCoV-229E), we do not propose that this would explain the reduced severity of COVID-19 observed in patients infected with Omicron compared to prior VOCs [81,82,83,84].

Multiple studies have demonstrated reduced effectiveness of COVID-19 vaccines against the Omicron variant compared to prior variants, including both primary vaccination series and booster doses [85,86,87,88,89]. Mechanistic characterization of the immune-evasive properties of Omicron have highlighted several substitutions in the Spike protein that confer antibody resistance, but have not revealed a role for ins214EPE [69,90,91,92]. The absence of this insertion in Omicron sublineages that emerged after BA.1 (Appendix A) raises the question of whether this insertion was critical for the initial rapid global spread of Omicron, or if it was rather acquired as a non-essential “passenger mutation” during its evolution. It is also important to recognize that throughout the pandemic, different SARS-CoV-2 variants have variably impacted populations in distinct regions around the globe. For example, the Gamma variant spread predominantly in South America, while the Beta variant was more prominent in South Africa, Europe, and Asia. Omicron has demonstrated the capacity to spread globally, but whether and how prior exposure to these different variants impacts its transmission is worth further exploration. Finally, the lack of clinical annotation associated with publicly deposited viral genomic data limits our ability to assess how the trajectory of viral evolution (including the acquisition of ins214EPE and other mutations) may be impacted by features such as vaccination status and immune competence. Future analyses of clinically annotated viral samples could help to address these questions.

## 5. Conclusions

The rapid rise in COVID-19 cases attributed to the Omicron variant, including among fully vaccinated individuals, raised alarm globally. In this context, it is important to better understand both the origins and the consequences of new genomic alterations that distinguish Omicron from prior VOCs and VOIs. Here, we begin to address the former by providing several plausible hypotheses on the origins of a nine-nucleotide insertion in the N-terminal domain of the Spike protein of the initially identified Omicron BA.1 variant. We suggest that genomic surveillance strategies should include an emphasis on sequencing SARS-CoV-2 genomes from immunocompromised patients and individuals with viral co-infections (including co-infections with multiple SARS-CoV-2 variants), as such individuals may provide unique contexts for genomic recombination and the evolution of new variants.

## Figures and Tables

**Figure 1 vaccines-10-01509-f001:**
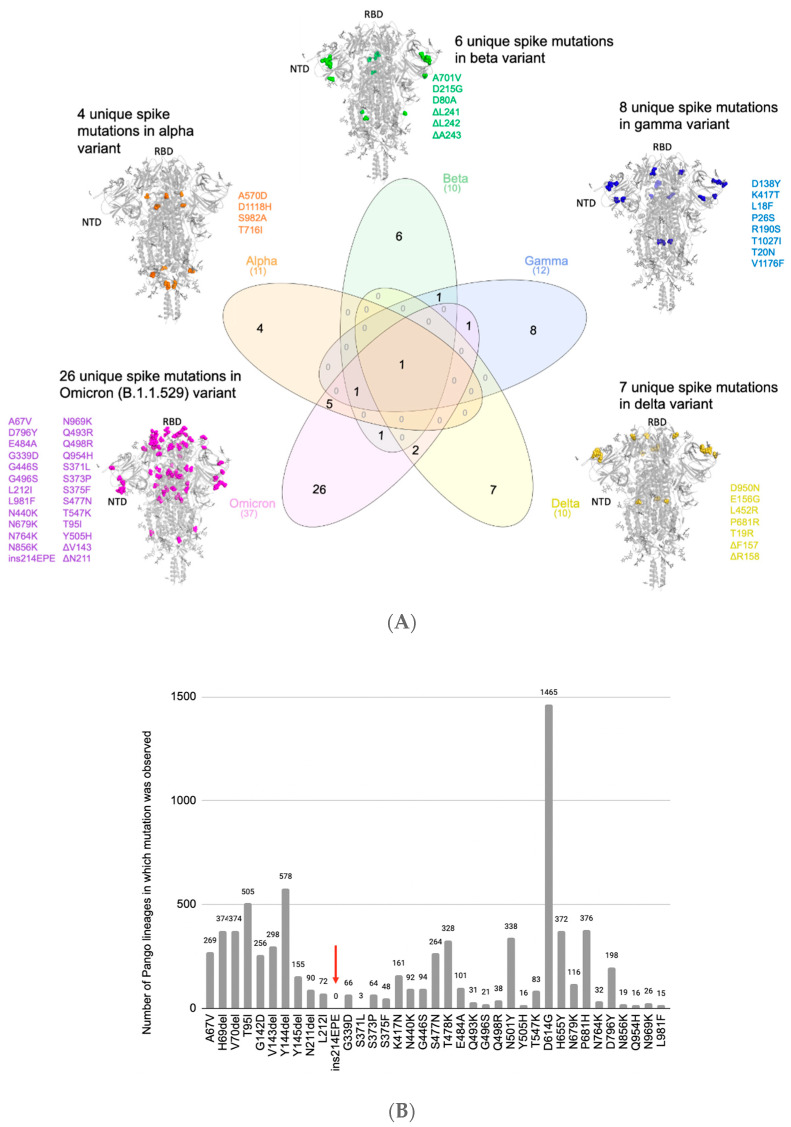
Specificity of Omicron’s Spike mutations relative to SARS-CoV-2 variants of concern and other lineages. (**A**) Comparing the lineage-specific Spike protein mutations in the SARS-CoV-2 variants of concern. The unique mutations observed in the Spike protein for each of the variants are highlighted (spheres) on the homotrimeric Spike protein of SARS-CoV-2. The Omicron (B.1.1.529/BA.1/BA.2) variant has the highest number (26) of unique mutations in the Spike protein from this perspective, making its emergence a “step function” in the evolution of SARS-CoV-2 strains. (**B**) Prevalence of Omicron’s Spike mutations in other SARS-CoV-2 lineages. Bar plot denoting the number of SARS-CoV-2 lineages (besides Omicron) in which the mutations present in Omicron are observed. The red arrow highlights the insEPE214 mutation, which is absent from all other SARS-CoV-2 lineages.

**Figure 2 vaccines-10-01509-f002:**
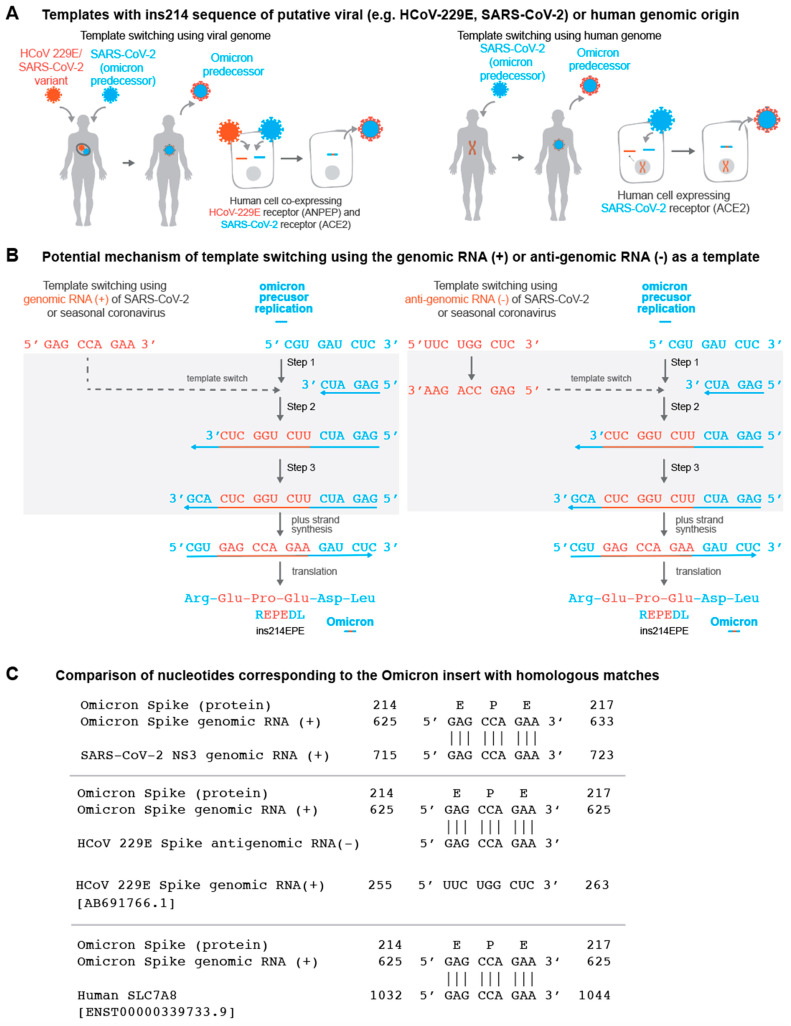
(**A**) Schematic representation of Omicron’s evolution through template switching involving viral (e.g., seasonal coronavirus or SARS-CoV-2) or human RNA. (**B**) Potential mechanism of template switching using viral genomic RNA (positive sense) or anti-genomic RNA (negative sense) as a template. Step 1: Negative-strand synthesis begins using an Omicron predecessor’s genomic RNA as template. Step 2: Negative-strand synthesis temporarily uses the genomic or anti-genomic RNA of SARS-CoV-2 or a co-infecting virus. Step 3: Negative-strand synthesis resumes using the Omicron predecessor’s genomic RNA as template. (**C**) Examples of matches identical to the nucleotide sequence “GAG CCA GAA” in the SARS-CoV-2 genome, the HCoV-229E anti-genome, and a human SLC7A8 transcript are shown.

**Table 1 vaccines-10-01509-t001:** Numbers of viral genomes and human transcripts with forward or reverse-complement matches to the three potential insertion sequences. Because the insertion sequences occur by definition in Omicron genomes, counts are shown separately for total SARS-CoV-2 genomes from GISAID and for genomes that are not assigned to the Omicron lineage. We confirmed that no sequences that were assigned to the Omicron lineage contained the insertion sequence or its reverse complement at any sites other than at the insertion itself. For the human *Coronaviridae* genomes, counts were first obtained by considering the available sequences for all human-infecting coronaviruses, and then filtered to retain only the viruses that are known to cause common colds or enteric illness (i.e., severe acute respiratory syndrome and Middle East respiratory syndrome virus sequences were excluded).

Candidate Insertion Sequence	Human Transcriptome		SARS-CoV-2 Genomes from GISAID		Human *Coronaviridae* Genomes	
	Transcripts (Genes)		Total (Non-Omicron)		Any (Seasonal or Enteric)	
	Forward	Reverse-Complement	Forward	Reverse-Complement	Forward	Reverse-Complement
5′-GAGCCAGAA-3′	4677 (1534)	3264 (1220)	2100 (931)	27 (27)	18 (8)	17 (6)
5′-AGCCAGAAG-3′	6190 (2008)	4293 (1591)	1275 (106)	269 (269)	3 (0)	12 (0)
5′-GCCAGAAGA-3′	5210 (1564)	3146 (1144)	1319 (150)	201,632 (201,632)	13 (7)	4 (2)

## Data Availability

The data analyzed in this study was obtained from publicly available resources including GISAID (https://www.gisaid.org), CoV-RDB (covdb.standford.edu), and the nferX Single Cell Platform (https://academia.nferx.com/dv/202011/singlecell/).

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
