# Peer review of "On the Origins of Omicron’s Unique Spike Gene Insertion"

_vaccines, 2022, doi:10.3390/vaccines10091509_

Round 1

Reviewer 1 Report

I find the presented manuscript to be accurate and scientifically sound.

I see no faults in the way data is presented in the current version of manuscript, and only have some minor suggestions:

(1)   Page 2, line 52: furin instead of FURIN

(2)   Page 7 line 166: template switching

(3)   Maybe it makes more sense to put table 1 in Supplements

(4)   Page 11, lines 283-284: it makes sense to put description of your random dataset in Methods

(5)   In discussion while discussing Omicron origin you mention non-human host but nothing about the immunocompromised patients, while you state in the end of this section that one must monitor those patients. It makes sense to mention immunocompromised patients at line 324 on page 12.

Reviewer 2 Report

In this manuscript, Venkatakrishnan et al. present a model suggesting several possible mechanistic origins of a spike insertion.  Though this is conceptually intriguing and has potential implications for the landscape of future mutations if true, I do not think the authors were able to make a definitive case that any of the candidate source templates was, in fact, the one that was used. They were unable to show any flanking regions of homology, suggesting the possibility that sequence homology may not be required but also leaving open the possibility that they may not have identified the template that was actually used or that a different mechanism was involved.

Line 24-25: I agree that it is important to know whether ins214 EPE impacts epidemiological or clinical properties and whether there is “inter-viral or host-virus genomic interplay” but they do not provide any direct evidence for either of these and thus, the concluding sentences of the abstract are somewhat misleading.

Line 38: It is now greater than 6 months after the sequences data was collected with new omicron sub-lineages emerging so would be valuable to update.

Line 76: It is not clear to me how certain mutations in omicron are surge associated while others are not if they are all characteristic of the omicron lineage.

Line 95, 110: Please indicate which insertions have been located near this position in the table.

Line 101-102: The authors state that all of the other substitution and deletion mutations have been found in previous SARS-CoV-2 lineages.  I do not follow this statement as there seem to be multiple point mutations (L212I, G339D, S317L, etc) and deletions (N211del) that are not listed in other variants. Do they mean to indicate that these are minor populations within the lineage?  If so, perhaps this could be indicated on the table?

Line 110: If other SARS-CoV-2 lineages have acquired insertions, this would be a useful comparison.  A 8-amino acid insertion would give a longer stretch of nucleotides to map against sequence databases to identify potential template sequences.   Similarly, are there other examples of insertions within other SARS-CoV-2 genes that might suggest use of a common template for switching, for example sequence insertions that map to other coronavirus sequences.

Line 111: It would help to understand this if the insertions from other lineages could be illustrated in the table.

Line 111: I don’t know that I would characterize a single example in the literature as “well-established potential.”

Line 159: If longer insertions can be generated by local duplications, it raises the question as to whether the ins214 might have originally arisen as a duplication with subsequent point mutations obscuring the similarity to the parental sequence?

Line 184: Could other RNA respiratory viruses (e.g. influenza, respiratory syncytial virus, parainfluenza virus, human metapneumovirus) also be considers as possible sources of template RNA?

Line 284: This statement is predicated on the assumption that one of the host or viral mRNAs served as the template.  I do not think the authors have demonstrated this fact conclusively.

Line 367: It is now >6 months since most recent data retrieval, it would be desirable to include some addition sequences to characterize the expanding omicron lineage.

Line 384: Could it be that the insertion changed in an ancestral virus such that the current sequence is no longer the same as the originally inserted sequence?  The methods indicate that they allowed one nucleotide mismatch but might it be possible that the sequenced changed more from the putative ancestral insertion, given the large number of substitutions identified in the omicron lineage?

In the absence of any surrounding sequence homology, can the authors suggest a mechanism by which this particular sequence was selected over the other cellular mRNAs and viral sequences.  If template switching were common, one might expect to see more insertions.

Reviewer 3 Report

The study performed by Venkatakrishnan is quite interesting that highlights one of the vital feature of this pandemic that foresee the variant of existing wild type SARS-CoV-2. The authors have delved deeper into to explore the key mutations in Omicron variant and how this may impact in general. The study in well made and is timely. However i am satisfied with most the of the work, i still have few questions to ask in the context of current study.

1. How these mutation have impacted on vaccine efficacy?

2. As numerous mutation of SARS-COV-2 has led to different variants. These variants have impacted different populations (eg. Asian, African) differently. Means at one end they are quite devastating and have negligible influence  in another population. How authors observe this through their findings.

3. How about considering the population that have not been vaccinated so far (vaccine hesitancy). Do such individuals sequences provides a newer story and may provide scientific evidence to vaccine hesitancy claiming the role of vaccines in such frequent mutations of SARS-COV-2.
